# The Stability Guided Multidisciplinary Treatment of Skeletal Class III Malocclusion Involving Impacted Canines and Thin Periodontal Biotype: A Case Report with Eight-Year Follow-Up

**DOI:** 10.3390/medicina58111588

**Published:** 2022-11-03

**Authors:** Juan Li, Xiaoyan Feng, Yi Lin, Jun Lin

**Affiliations:** Department of Stomatology, The First Affiliated Hospital, College of Medicine, Zhejiang University, Hangzhou 310003, China

**Keywords:** skeletal class III malocclusion, connective tissue graft, orthognathic surgery, impacted maxillary canines

## Abstract

Skeletal class III malocclusion with severe skeletal disharmonies and arch discrepancies is usually treated via the conventional orthodontic-surgical approach. However, when associated with tooth impaction and periodontal risks, the treatment is more challenging and complex. The esthetic, occlusal, and periodontal stability of the treatment outcome is more difficult to obtain. The 16-year-old female patient in this case was diagnosed with dental and skeletal Class III malocclusion, bilateral impacted maxillary canines, and scalloped thin gingiva. The multidisciplinary management included a segmental arch technique, extracting two premolars, a subepithelial connective tissue graft surgery, and orthognathic surgery. The esthetic facial profile, pleasant smile, appropriate occlusion, and functional treatment results were obtained and maintained in 8-year follow-up.

## 1. Introduction

Skeletal class III malocclusion has a high rate of prevalence in Asian people, and it has a high rate of relapse following orthodontic treatment, which poses a challenge to orthodontists [1]. Combined orthodontic and orthognathic surgery is a conventional option to correct the malocclusion and dentofacial deformities in adults with severe skeletal class III malocclusion.

In this type of patient, hypodevelopment of the maxilla is common, and an insufficient length and width of the maxilla can lead to maxillary teeth impaction, of which canines are the most frequent [2]. Maxillary canine impaction occurs in approximately 2% of the population, while the incidence in the maxilla is more than twice that in the mandible, and only 17% of labially impacted canines have enough space [3]. The various treatment options available are observation, intervention, relocation, and extraction. Orthodontic traction is a widely used and efficient method to reserve and position the canine in its proper location within the arch; however, there are difficulties and considerations since it may cause pulp necrosis, gingival recession, or alveolar-dental ankylosis [4].

In addition, the periodontal condition has a significant impact on the limitation of orthodontic tooth movement, macro and micro aesthetics, and stability [5]. In patients with skeletal class III malocclusion, the gingival thickness in the area is also found to be thinner, with a so-called thin scalloped gingival biotype accompanied by relatively deficient underlying bone and attached gingiva [6]. The teeth in the mandibular anterior area is moved labially on the narrow alveolus in the pre-surgery decompensation period [7]. These all indicate that the risk of gingival recession, fenestrations, and dehiscence is highly elevated. Dealing with the corresponding periodontal risks and maintaining stability are also huge challenges.

The treatment for skeletal class III malocclusion patients with both impacted canines and thin periodontal biotype is more complicated in plan elaborating and retention designing. This clinical report provides an interdisciplinary treatment strategy through a typical case study. In the case, a segmental arch technique, a subepithelial connective tissue graft surgery, and an orthognathic surgery were performed, and favorable esthetic and stable occlusion were obtained and maintained in follow-up period of 8 years.

## 2. Case Report

### 2.1. Diagnosis and Etiology

The patient, a 16-year-old female with no significant medical history, presented to the Orthodontic Department seeking to correct occlusion and improve her face aesthetic.

The facial examination displayed a concave profile, a prominent chin, an increased lower third of the face, and an unconfident smile. The upper lip was retruded 4.8 mm in relation to the E plane (Figure 1). The intraoral photographs (Figure 1) and dental casts (Figure 2) showed a bilateral Class III molar relationship and an anterior crossbite with a negative overjet of 2 mm. The width of the maxilla was narrow compared to the mandible, which led to a crossbite in the right posterior region and a compensatory lingual inclination in the left mandibular posterior region. All the mandibular deciduous molars and maxillary deciduous canines were retained with the left permanent maxillary canine erupted labially. Scalloped thin gingiva was evident in the mandibular anterior region with an obvious root shape. Temporomandibular disorder symptoms or bad oral habits were not detected. The mandible cannot retreat to the edge-to-edge occlusion.

The panoramic radiograph showed that both the maxillary canines and all the third molars were impacted, and no significant periodontal support loss was found. The cephalometric analysis (Figure 3 and Table 1) showed a severe skeletal Class III relationship (ANB, −4.0°) with an insufficient developed maxilla (SNA, 77.2°). The maxillary incisors were relatively well-positioned, while the mandibular incisors were lingually inclined (U1-SN,104°; L1-MP, 86.5°) [8].

### 2.2. Treatment Objectives

The treatment objectives were to: (1) tract the impacted maxillary permanent canines; (2) improve the periodontal phenotype of the lower anterior region and reduce the risk of gingival recession through mucogingival surgery; (3) solve the horizontal and sagittal discrepancy between the mandible and the maxilla to improve the facial profile through two-jaw surgery; and (4) align the dentition to establish function occlusion.

### 2.3. Treatment Alternatives

Little or no orthopedic maxillary and mandible response could be expected because the female patient was already 16 years old with little growth potential. Therefore, orthognathic surgery could be a proper choice to solve her complaints. Two-jaw surgery includes LeFort I osteotomy for maxilla advancement and bilateral split sagittal osteotomy for mandible setback because the stability of isolated mandibular setback is relatively poor.

As for the dental problem, the first option was the extraction of two impacted maxillary canines to shorten the treatment time and offer the spaces to decompensate and retract the maxillary incisors. However, the maxillary canines were vital in dentition, as they frame the smile and guide occlusion [9]. The second option was the removal of the maxillary first premolars, which would resolve the crowding of the anterior area and regain space for traction. This approach would contribute to an extended course of presurgical treatment. However, according to her age, the timing of orthognathic surgery was not yet appropriate, so the whole treatment course would not be extended. Considering the periodontal conditions described above, mucogingival surgery was arranged to improve the periodontal tissue quality and the treatment outcome [10].

### 2.4. Treatment Progress

Before the orthodontic treatment, the extraction of all retained deciduous teeth and maxillary first premolars were scheduled. After the extraction, both arches were bonded with preadjusted brackets (0.022-inch slot; 3M Unitek, Monrovia, CA, USA), aligned with initial 0.014-inch nickel titanium arch wires, and changed sequentially to eliminate crowding and provide leveling. Meanwhile, two impacted maxillary permanent canines were tracted through two auxiliary segmental 0.019 × 0.025-inch stainless steel arch wires with vertical helical loops (Figure 4). After 12 months, the canines were basically tracted into the right place. Presurgical decompensation started to increase the magnitude of surgical movement. The unfavorable tooth inclinations were corrected with 0.019 × 0.025-inch stainless steel arch wires through sliding mechanics to increase reverse overjet. A subepithelial connective tissue graft was also performed. Two connective tissue grafts were harvested from the palate and positioned in the prepared recipient site corresponding to the mandibular anterior region. The attached gingiva and keratinized tissue were augmented after the mucogingival surgery (Figure 5). After 34 months, the dentition preparation phase of presurgical treatment was completed in both arches (Figure 6).

Based on the reconstructed data of CBCT(cone-beam computed tomography) and cast surgery, the orthognathic surgery was determined to require a LeFort I maxillary osteotomy to advance the maxillary in 5 mm and the bilateral sagittal split osteotomy to setback the mandible in 3 mm. During the surgery, two splints were subsequently applied to assist in accurately placing and maintaining the jaw’s position. Then, the new position was fixed with rigid internal fixation (RIF) and intermaxillary elastics. The patient was monitored closely after the procedure, and she was also taught how to perform opening and lateral movement exercises.

One month after the surgery, the spaces in the upper arch were closed with 0.019 × 0.025-inch stainless steel arch wires with a double key loop. At the finishing stage, a fine adjustment of the occlusion was applied to improve the anterior overjet, the overbite, and the canine and molar relationship. After a total treatment of 48 months, the multibracket system and all Micro-Implant Anchorages were removed. Lingual bonded retainers from canine to canine and Hawley retainers were placed immediately after removal.

### 2.5. Treatment Results

All the initial treatment objectives, including occlusion, periodontal health, and facial esthetics, were achieved by a satisfactory multidisciplinary approach, partly due to the cooperation of the patient. The facial photographs showed a pleasant profile and a harmonious smile. The patient was satisfied with the facial improvement, and she became more confident (Figure 7). Posttreatment intraoral photographs and dental casts (Figure 8) showed bilateral Class II molar and Class I canine relationships with an ideal overjet and overbite. The gingiva tissue in the mandibular anterior region was evidently augmented, which indicated lower periodontal risks.

The final panoramic radiograph confirmed parallel roots with no apparent root resorption. Cephalometric analysis (Figure 9 and Table 1) indicated a normal anteroposterior (AP) relationship (ANB, from −4° to 1.3°) and decreased lower third (FMA: from 26.6° to21.5°; Na-Me: from 111.8 mm to 103.7 mm). Furthermore, the distance from the upper and lower lips to the E-line were significantly decreased, which helped improve the soft tissue profile.

The 8-year follow-up photographs showed excellent stability of the occlusion and the profile (Figure 10). The pretreatment, posttreatment, and follow-up cephalometric superimposition revealed a significant improvement and stability in the facial profile and the skeletal and dental relationship. Superimposition of the posttreatment and retention digital dental models indicated generally stable results (Figure 11).

## 3. Discussion

Treatment plans for nongrowing patients with skeletal class III malocclusion are varied, including sole orthodontic treatment for dental camouflage or combined orthodontic-orthognathic treatment, including single or double-jaw surgery and genioplasty [11]. Johnston et al. reported that bimaxillary surgery was 3.4 times more likely to fully correct the ANB angle than mandibular surgery [12]. Previous studies have shown that maxillary advancement was stable, while large mandibular setback was a risk factor for relapse due to its habitual mandibular forward movement and prolonged growth. Double-jaw surgery, including bilateral sagittal split osteotomy and maxillary advancement, is the most effective way to improve stability [13]. Considering the degree of skeletal discrepancy in three dimensions in this case (ANB: −5.2°; Wits appraisal: −11.3 mm; FMA: from 26.6°; Na-Me 111.8 mm), the conventional surgical approach was determined to advance maxilla in 5 mm and setback mandible in 3 mm.

In addition to the maxilla-mandible relationship, the arch and the dentition play an important role in long-term stability. The etiology of the impacted teeth was related to the arch-length deficiency, which occurs in hypo-developed maxilla and may lead to bilateral impacted canines, which happened in this case. A treatment alternative is extraction; however, after extraction, there is a need to figure out an approach to replace the pivotal esthetic and the occlusion function of the missing canines, such as implant-retained crown restoration, conventional bridge, or premolar substitution through orthodontic treatment. The panoramic radiograph showed the canine crown was in the buccal side and distal to the midline of the lateral incisor, which noted a higher rate of successful traction, even up to 91% [3]. However, it takes space to decompensate and tract maxillary anterior teeth, so the extraction project was beneficial to create space [14]. Meanwhile, extraction guided the posterior dental arch to move relatively forward, which was equivalent to increasing the arch of the upper jaw and conducive to the establishment of a normal overjet and overbite of the posterior teeth. All things considered, a removal of the deciduous canines and the first premolars was decided, and then surgical exposure and a segmental arch technique were performed to tract the canines (Figure 12). The pre-surgical planning was mainly to tract the impacted teeth and create enough overjet and overbite for jaw movement, which was established a stable jaw–tooth relationship and improved the profile. The pre-surgical treatment went on for 2 years to achieve the traction of canines and reach the proper timing for the orthognathic surgery, which seemed longer compared to routine cases. Finally, we placed the canines correctly and established an ideal occlusion. Proper overjet and overbite, coordinate width, and close occlusion helped maintain the stable dentition relationship and further ensured the stability of the jaw’s position and soft tissue aesthetic.

Dental decompensation during presurgical orthodontic treatment is aimed to correct the torque of anterior teeth [15]. It is vital to keep an eye on the periodontal condition because mandibular anterior teeth always express crown lingual torque in skeletal class III malocclusion, which enhances the risks of gingival recession, fenestrations, and dehiscence [16]. Mucogingival surgery is a widely used technique to obtain attached gingiva, including free gingival graft, laterally positioned flap, and subepithelial connective tissue graft [17]. According to the thin periodontal biotype, we chose to use a subepithelial connective tissue graft, which was the best way to convert a thin, soft tissue to a thick biotype. Occlusal trauma caused by anterior crossbite was relieved by acquiring proper anterior overjet [18]. The outcome displayed that the keratinized and attached gingiva gained sufficient thickness, which preserved healthy a periodontal state and better represented the pink-white esthetic.

Retention and stability following an orthodontic-surgical approach is an important indicator to judge whether a case is successful. The 8-year follow-up showed a stable effect in hard or soft tissues in this case. We applied a rigid internal fixation to settle the jaws during the active orthodontic period, and then we applied a lingual bonded retainer and a Hawley retainer for both arches during the retention period [19]. In addition, the adaptable finishing occlusion and TMJ position were indispensable factors for long-term stability in our case.

In our treatment, digital technology was applied to visualize and design the treatment plan. Dolphin software was used to simulate the jaw movement and carry out a visual surgical design to clarify the target position (Figure 13). However, digital technology was not applied to design and manufacture the surgery splints. Virtual surgical planning with CAD/CAM technology [20] and artificial intelligence has been gaining popularity in orthognathic surgery. It can provide doctors and patients with more predictive treatment outcomes, and it can also contribute to more accurate surgical processes and more stable results [21].

## 4. Conclusions

Due to the multidisciplinary treatment, including the orthognathic surgery, the orthodontic treatment, and the periodontal procedures, the skeletal and dental class III relationship was corrected, the impacted canines returned to a normal position, and the periodontal condition was in a relatively stable state. Additionally, during the 8-year follow-up, the facial esthetics, functional occlusion, and periodontal health were successfully maintained. A stability-guided, close interdisciplinary approach is critical for successful and stable outcomes in similar cases.

## Figures and Tables

**Figure 1 medicina-58-01588-f001:**
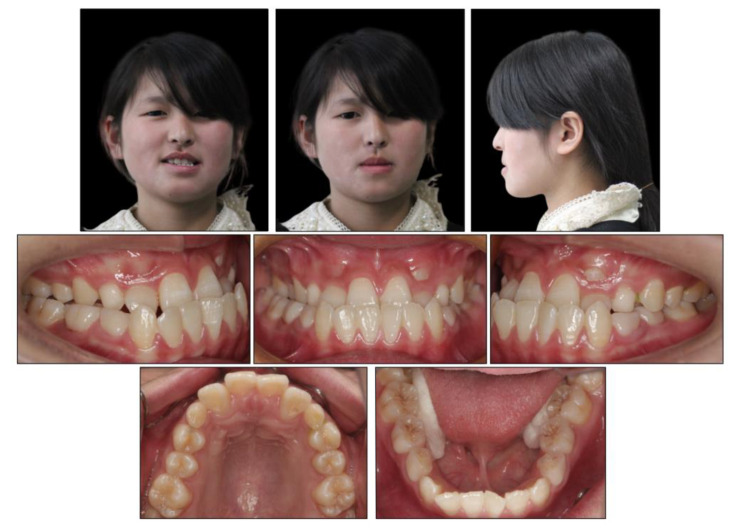
Pretreatment intraoral and facial photographs.

**Figure 2 medicina-58-01588-f002:**
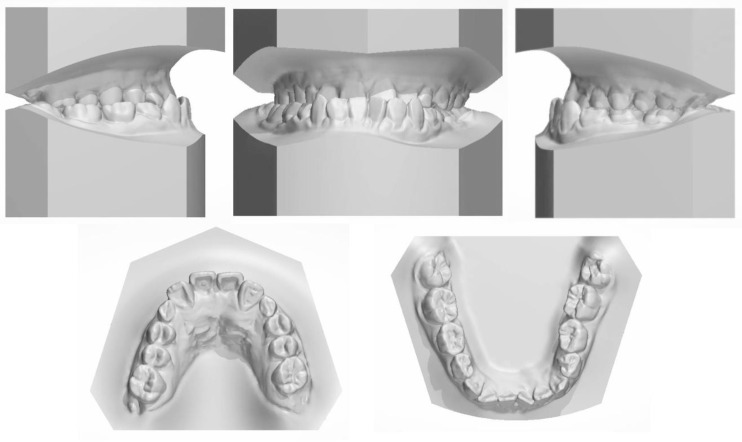
The pretreatment dental cast.

**Figure 3 medicina-58-01588-f003:**
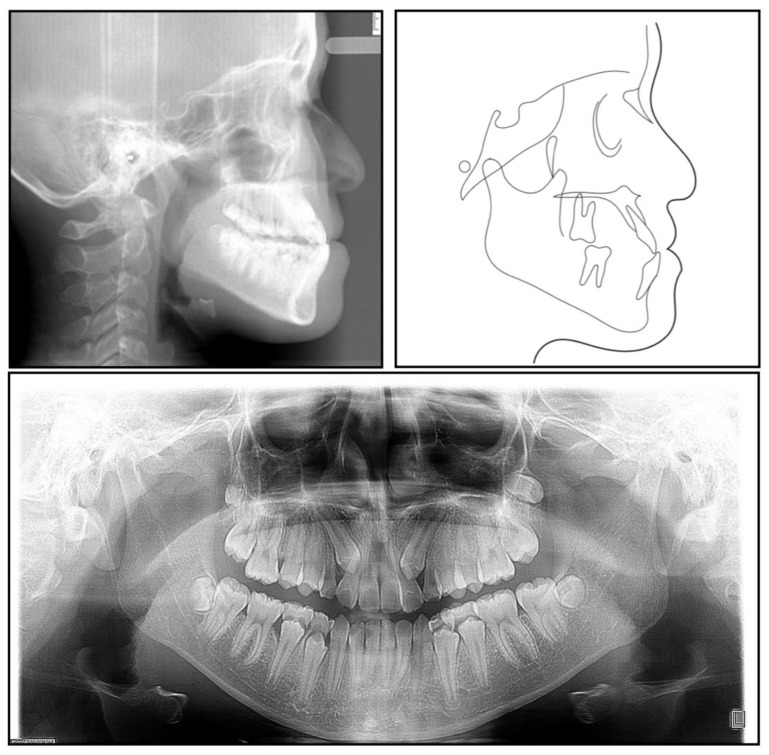
A pretreatment lateral cephalometric radiograph and tracing and a panoramic radiograph.

**Figure 4 medicina-58-01588-f004:**
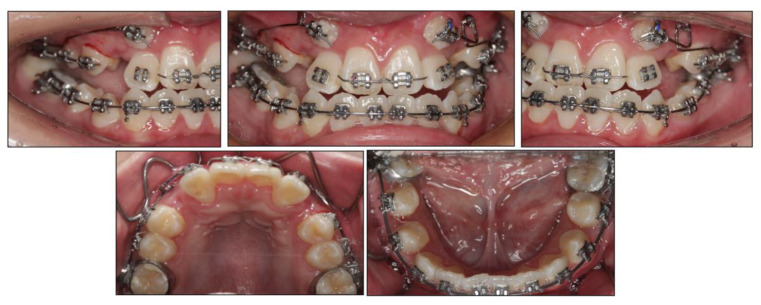
Tracting the impacted maxillary permanent canines with the segmental arch technique.

**Figure 5 medicina-58-01588-f005:**
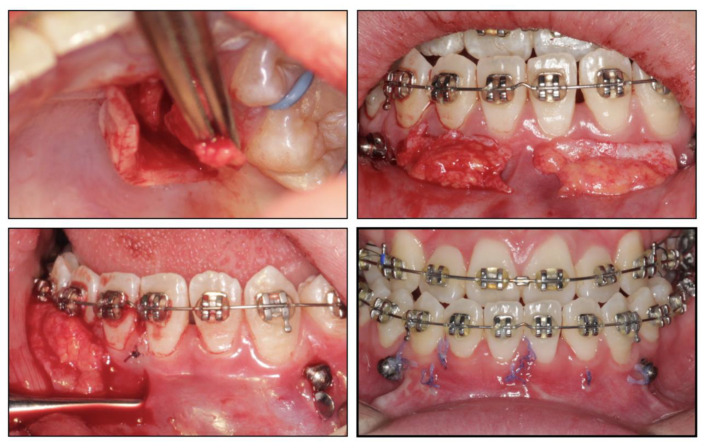
Subepithelial connective tissue transplantation.

**Figure 6 medicina-58-01588-f006:**
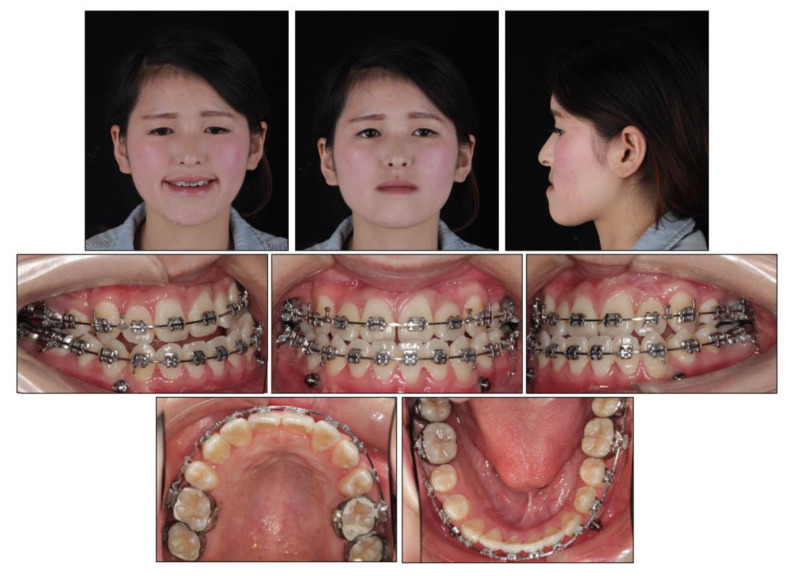
Presurgical facial and intraoral photographs.

**Figure 7 medicina-58-01588-f007:**
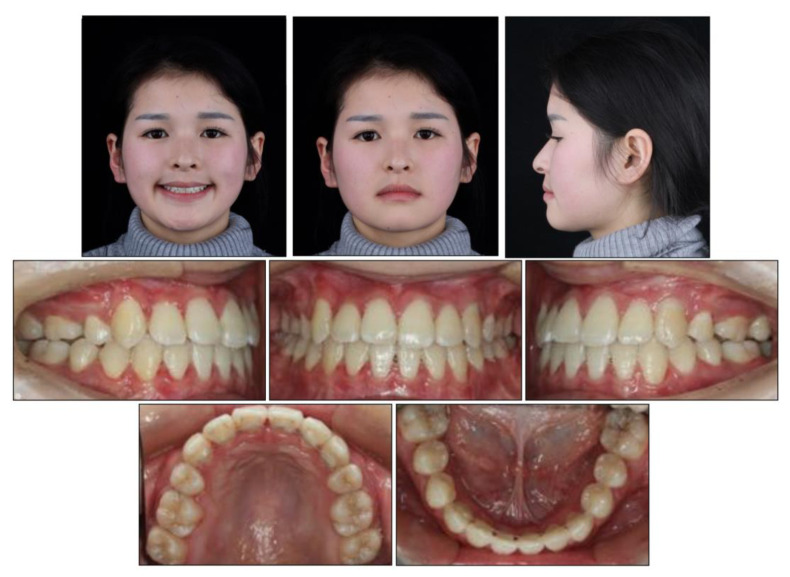
Posttreatment facial and intraoral photographs.

**Figure 8 medicina-58-01588-f008:**
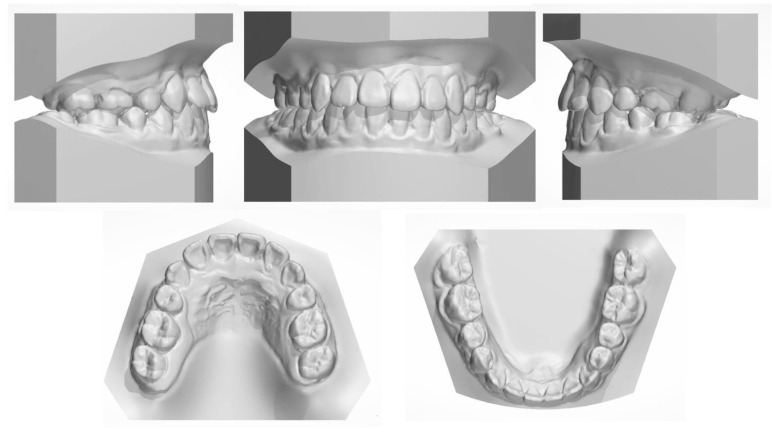
Posttreatment dental casts.

**Figure 9 medicina-58-01588-f009:**
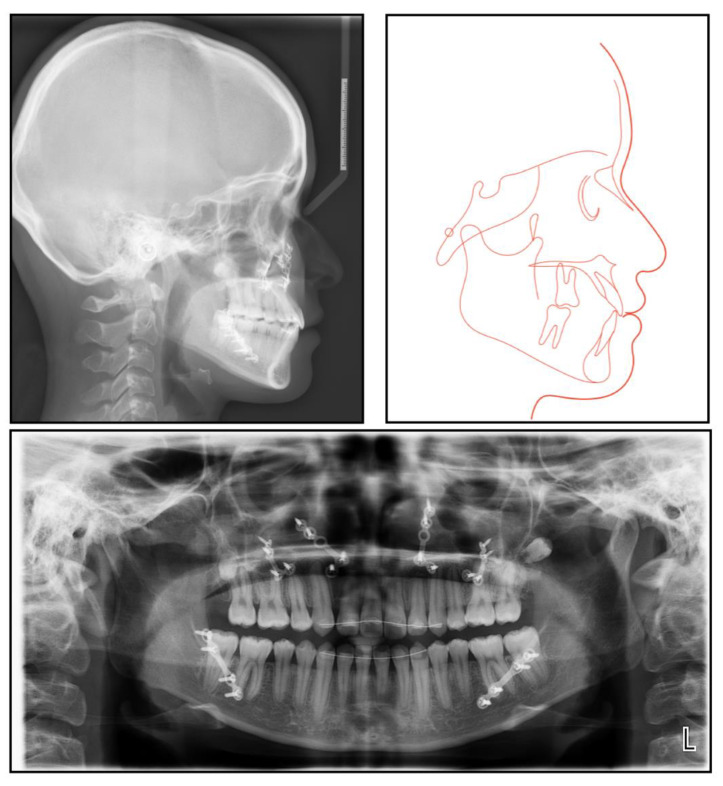
Posttreatment lateral cephalometric radiograph and tracing and a panoramic radiograph.

**Figure 10 medicina-58-01588-f010:**
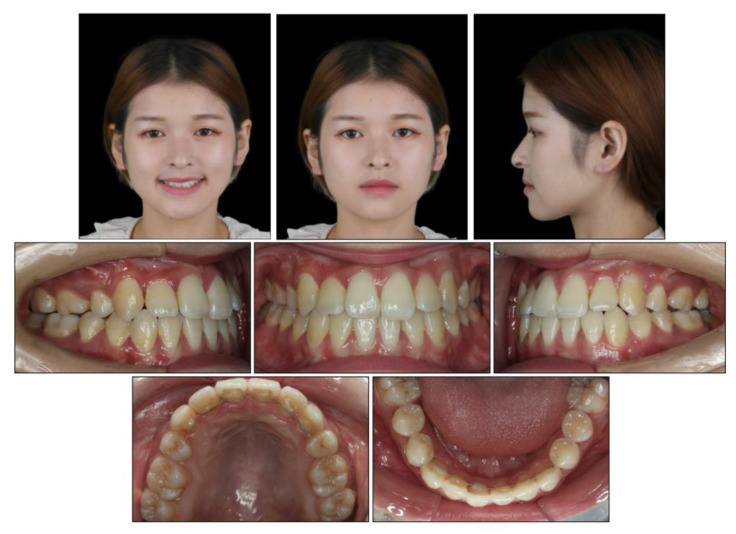
Facial and intraoral photographs after the 8-year follow-up.

**Figure 11 medicina-58-01588-f011:**
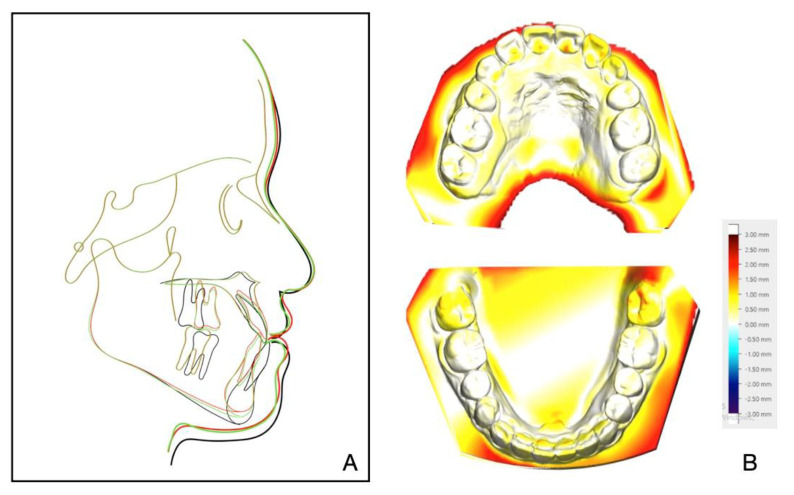
(**A**) Superimposed cephalometric tracings: pretreatment (black), posttreatment (red), and 8-year follow-up (green). (**B**) Superimposition of the posttreatment and retention digital dental models.

**Figure 12 medicina-58-01588-f012:**
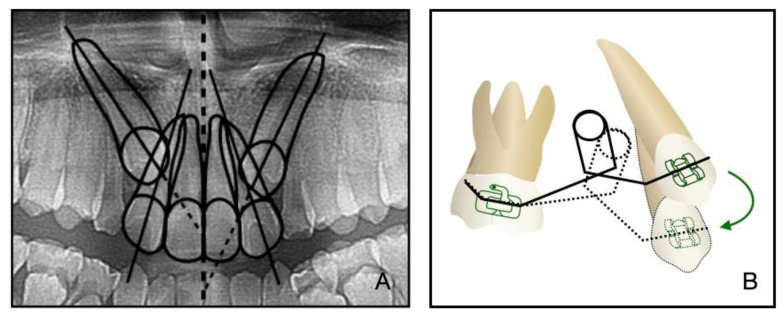
(**A**): The relative position of the canine crown and lateral incisor in a panoramic radiograph. (**B**): Biomechanism of canine traction through segmental arch. The green arrow illustrates the path of canine movement.

**Figure 13 medicina-58-01588-f013:**
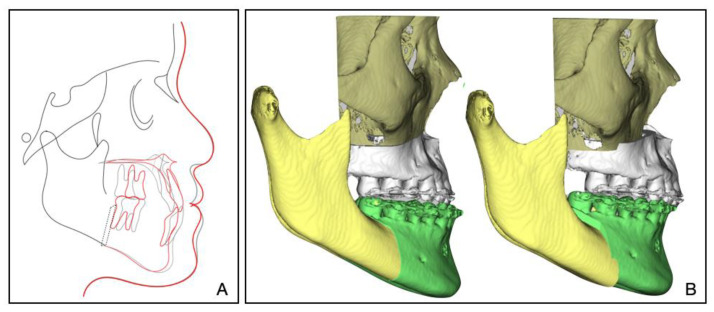
(**A**) VTO (Visual Treatment Objective): The postoperative effects were simulated by lateral radiographs. Pretreatment (black) and post-treatment (red). (**B**) Digital orthognathic surgery design through Dolphin software.

**Table 1 medicina-58-01588-t001:** Cephalometric measurements.

Measurement	Norm ± SD	Pretreatment	Posttreatment	8-Year Follow-Up
FMIA (°)	64.8 ± 8.5	67.6	69.0	68.0
FMA (°)	23.9 ± 4.5	26.6	21.5	22.6
SNA (°)	82.0 ± 3.5	77.2	83.0	83.4
SNB (°)	80.9 ± 3.4	81.3	81.6	81.5
ANB (°)	1.6 ± 1.5	−4.0	1.3	1.9
L1-MP (°)	95.0 ± 7.0	86.5	89.6	89.3
Occ-Plane (°)	6.8 ± 5.0	8.1	9.5	10.7
U1-SN (°)	102.8 ± 5.5	104.0	105.6	103.5
Upper Lip-E (mm)	6.0 ± 2.0	−4.8	−4.0	−4.2
Lower Lip-E (mm)	−2.0 ± 2.0	−0.3	−1.5	−1.1
S-Go (mm)	82.5 ± 5.0	67.9	67.8	68.1
Na-Me (mm)	128.5 ± 5.0	111.8	103.7	105.4
S-Go/Na-Me (%)	65.0 ± 4.0	60.7	65.4	64.6

S, sella; N, nasion; MP, mandibular plane; Occ-Plane, occlusion plane; E, aesthetic line; Go, Gonion; Me, Menton.

## Data Availability

All experimental data supporting the results of this study are available from The First Affiliated Hospital, College of Medicine, Zhejiang University.

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
