# Peer review of "The Stability Guided Multidisciplinary Treatment of Skeletal Class III Malocclusion Involving Impacted Canines and Thin Periodontal Biotype: A Case Report with Eight-Year Follow-Up"

_medicina, 2022, doi:10.3390/medicina58111588_

Round 1

Reviewer 1 Report

The read this manuscript with great interest, and the case report under review is well presented. All the sections of the manuscript are very well written by the authors and need a few changes as mentioned below

Introduction:

1.       Combined orthodontic and orthognathic surgery is a conventional option to correct the malocclusion and dentofacial deformities for adults with severe skeletal class III malocclusion: kindly provide a reference

2.       Kindly write about the rationale, refer to and cite accordingly the below link

https://link.springer.com/chapter/10.1007/978-981-15-1346-6_66

Case report:

1.       Suggest adding treatment planning

2.     Pre surgical planning and digital technology utilization, provide details

3.       Kindly provide illustrations to make it reader easy.

Reviewer 2 Report

Thank you for the trustworthy and honourable task of reviewing this manuscript. After a meticulous review of the manuscript, I found that the submitted case report meets the requirements and high quality standards of medicina and might therefore interersting for its reades.

The authors present a case report with eight years of follow-up. The treatment of a 16 year old female patient diagnosed with skeletal Class III malocclusion, bilateral impacted maxillary canines gives useful insight in the elaborate planning process of orthognathic surgery.

I congratulate the authors for their great longterm results.

Reviewer 3 Report

This case study offers and supports interdisciplinary care for stable, long-term therapeutic success, but further explanation is required, including:

- Please include the treatment plan, which must address the mucogingival and jaw surgery objectives in section 2.2.

- lines 57-58 "The facial examination displayed a concave profile, an increased lower third, a deficient smile, and a prominent chin." I assumed the authors mean the lower third of the face, it would be better understanding if the authors state clearly. Also, the deficient smile needs more explanation.   

- What reference of cephalometric analysis did you use?

- What is(are) the factor(s) considered to be the appropriate time for jaw surgery after teeth alignment?

- please check the word "bilateral split sagittal osteotomy", should it be  "bilateral sagittal split osteotomy"?

- What is the reason for leaving the impacted maxillary third molars?

- Her delayed eruption of mandibular premolars is interesting to discuss.
